Functional analysis of ARF1 from Cymbidium goeringii in IAA response during leaf development

Xu Zihan 1
Li Fangle 1
Li Meng 1
He Yuanhao 1
Chen Yue 2
Hu Fengrong 1 hufengrong2003@sina.com
1 College of Landscape Architecture, Nanjing Forestry University , Nanjing, Jiangsu , China
2 Institute of Horticulture, Zhejiang Academy of Agricultural Science , Hangzhou, Zhejiang , China
Mikami Koji
Electronic publication date: 2022 Mar 10
Publication date: 2022
Volume: 10
Electronic Location ID: e13077
Received 2021 Nov 18; Accepted 2022 Feb 16
Copyright: © 2022 Xu et al.
Copyright year: 2022
Copyright holder: Xu et al.
License: This is an open access article distributed under the terms of the Creative Commons Attribution License, which permits unrestricted use, distribution, reproduction and adaptation in any medium and for any purpose provided that it is properly attributed. For attribution, the original author(s), title, publication source (PeerJ) and either DOI or URL of the article must be cited.
License URL: https://creativecommons.org/licenses/by/4.0/

Keywords: ARF, Cymbidium goeringii, Auxin, Leaf development, Arabidopsis

Funding: National Natural Science Foundation of China 31801891 University Brand Major Construction Foundation of Jiangsu Province PPZY2015A063 Postgraduate Research & Practice Innovation Program of Jiangsu Province KYCX21_0928 This work was supported by the National Natural Science Foundation of China (No. 31801891), the University Brand Major Construction Foundation of Jiangsu Province (PPZY2015A063) and the Postgraduate Research & Practice Innovation Program of Jiangsu Province (KYCX21_0928). The funders had no role in study design, data collection and analysis, decision to publish, or preparation of the manuscript.

==============================
Background

Cymbidium is an economically important genus of flowering orchids cultivated in China because of showing graceful leaf shapes and elegant flower coloration. However, the deterioration of the ecological environment and the difficulty of conservation management have become increasing challenges for maintaining its germplasm resources. ARFs are critical transcription factors in the auxin signaling pathway and have been found to play pivotal roles in leaf growth and development in previous studies. However, their functions and mechanisms in Cymbidium goeringii remain to be clarified.

Methods

The sequence of the CgARF1 gene was analyzed by bioinformatics. The expression of this gene in different tissues and under IAA treatment was detected by quantitative real-time PCR analysis. The CgARF1 gene was overexpressed in wild-type Arabidopsis and Nicotiana benthamiana via the Agrobacterium infection method. Acetone-ethanol solvent extraction was applied for the determination of chlorophyll contents, and the contents of endogenous hormones were determined using the enzyme-linked immunosorbent assay technique.

Results

CgARF1 cloned from C. goeringii ‘Songmei’ was 2,049 bp in length and encoded 682 amino acids containing three typical domains: a B3 DNA binding domain, an Aux_resp domain and an AUX/IXX family domain. The expression pattern of CgARF1 in different tissues of C. goeringii showed that its expression was highest in the leaves and changed greatly under IAA treatment. Subcellular localization studies showed that the protein was mainly localized in the cell nucleus. CgARF1-overexpressing lines exhibited leaf senescence and a decreased chlorophyll content. Under IAA treatment, CgARF1 regulates the rooting length, rooting number and rooting rate from detached leaves. The levels of endogenous hormones in transgenic leaves were also significantly changed.

Conclusion

These results indicated that CgARF1 overexpression is responsive to IAA treatment during leaf development. This study provides a foundation for future research on the function of the ARF gene family in C. goeringii.

Introduction

Auxin is a key hormone that plays a pivotal role in many processes, such as embryogenesis, vascular differentiation and organ development, throughout the plant life cycle (Dharmasiri & Estelle, 2004; Su et al., 2014). When plants are stimulated by the external environment, changes in the dynamics and distribution of auxin may be initiated or mediated by auxin-regulated gene expression (Vanneste & Friml, 2009; Guilfoyle & Hagen, 2007; Chandler, 2016). The indole-3-acetic acid (IAA)-mediated Transport Inhibitor Response 1 (TIR1)-Aux/IAA-Auxin Response Factor (ARF) pathway has been accepted as the canonical auxin signal transduction pathway (Kubes & Napier, 2018). As a critical factor in this pathway, ARF can interact with TGTCTC auxin-responsive elements (AuxREs) in promoter regions, thereby regulating the expression of auxin-mediated genes (Ulmasov, Hagen & Guilfoyle, 1999).

Our current knowledge about the ARF family has been obtained mainly from plants; 50, 25, 39, and 14 ARF members have been identified in rice, tobacco (Nicotiana tabacum), poplar (Populus trichocarpa) and Dendrobium officinale, respectively (Wang et al., 2007; Sun et al., 2016; Kalluri et al., 2007; Chen et al., 2017). The identification and functional analyses of ARF family members in numerous plants have provided important insights into the mechanisms underlying the regulation of ARFs in the auxin signaling pathway.

ARF proteins are generally reported to show a modular structure with three domains, including a DNA-binding domain (DBD), a middle region (MR) and a carboxy-terminal dimerization domain (CTD). The DBD is a conserved N-terminal domain that is not responsive to auxin but can target AuxREs independently. The second, MR, domain is a nonconserved region with varying lengths that has been proposed to function as a transcriptional repression or activation domain (Tiwari, Hagen & Guilfoyle, 2003). The third domain, the CTD, contains two motifs (designated III-IV), which are homologs to regions of Aux/IAA proteins and facilitate the formation of heterodimers among ARFs and Aux/IAAs (Remington et al., 2004; Dreher et al., 2006). However, all ARF proteins do contain all three complete domains, as is the case for AtARF3/13/17/23 in Arabidopsis (Guilfoyle & Hagen, 2001; Remington et al., 2004) and CiARF3/14/17 in sweet orange (Li et al., 2015).

Previous studies have shown that ARF genes are involved in leaf growth and development. In Arabidopsis, ARF1 and ARF2 function in the auxin-mediated regulation of leaf longevity, and arf1 mutation enhances many arf2 phenotypes. AtARF1 acts in a partially redundant manner with AtARF2 during leaf senescence (Ellis et al., 2005; Lim et al., 2010). arf3 (ett)/arf4 double mutants exhibit a rolled leaf phenotype (Pekker, Alvarez & Eshed, 2005), while double mutants of mp/arf5 with arf3 or arf7 show a breakdown of leaf formation (Schuetz, Fidanza & Mattson, 2019), and arf2/3/4 triple mutants show severe leaf margin defects (Guan et al., 2017), indicating that these AtARFs are related to leaf morphology and polarity development and show functional redundancy. AtARF3 also plays a role in the regulation of leaf developmental timing and patterning (Fahlgren et al., 2006).

In addition to Arabidopsis, the ARF family has been reported in some other plant species. A mutation in the SlARF12 gene results in abnormal early leaf development in tomato (Solanum lycopersicum) (Kumar, Tyagi & Sharma, 2011), and SlARF2 influences leaf senescence (Guan et al., 2018). Additional studies have shown that a Sl-miR160a-resistant form (mSlARF10) specifically inhibits leaflet blade outgrowth without affecting other auxin-driven processes during compound leaf development (Hendelman et al., 2012). In rice (Oryza sativa), OsARFs control the lamina inclination bioassay by regulating the level of brassinosteroid receptors, suggesting multilevel interactions between auxin and brassinosteroids (Sakamoto et al., 2012). All of the above studies confirmed that ARF transcription factors (TFs) play pivotal roles in leaf growth and development, but with some differences across species.

Cymbidium is an important genus of economic flowering orchids cultivated in China (Dupuy, Ford-Lioyd & Cribb, 1985; Zhu et al., 2015). Cymbidium goeringii shows gracefully shaped leaves and elegantly colored flowers, which have made it popular among consumers, and this species therefore possesses great ornamental and economic value (Liu et al., 2020). However, the excessive excavation and deterioration of the ecological environment of this species have become increasing challenges in the maintenance of the germplasm resources and living environment of C. goeringii in recent years (Tang et al., 2015). ARF transcription factors participate in various biological processes in higher plants, such as vegetative growth, which is the basis of plant growth and development. Plant leaves are one of the vital organs of plants and the principal site of photosynthesis, and a good status of plant vegetative growth and leaf development is crucial for reproductive success. In this study, the ARF gene CgARF1 was identified and cloned from the Songmei cultivar of C. goeringii. We also evaluated CgARF1 expression in different plant tissues and its response mechanism to exogenous IAA treatment, and we further found that this gene was involved in the regulation of auxin-mediated leaf development in transgenic Arabidopsis. At present, little is known about the expression pattern and functions of ARFs in orchids. For this reason, research on CgARFs is not only innovative but also of great significance for breeding and genetic improvement in Cymbidium.

Materials and Methods

Plant materials and treatments

Plant tissues were collected from 2-year-old C. goeringii ‘Songmei’ plants grown under the conditions of 70% relative humidity, a temperature of 22 °C and natural light. The C. goeringii samples used in this study came from the Institute of Horticulture, Zhejiang Academy of Agricultural Sciences, Hangzhou, China. After collection, the samples were frozen in liquid nitrogen and stored at −80 °C until use. For the exogenous auxin treatment, leaves of C. goeringii were sprayed until dripping with 10 μM IAA and were sampled at 0 h, 2 h, 4 h, 6 h, 1 2 h, 24 h and 48 h.

Arabidopsis with a Columbia ecotype (Col-0) background was used for gene overexpression. This plant material was grown at (22 ± 1) °C under 75% relative humidity and a 16 h day/8 h night photoperiod in an artificial climate chamber. N. benthamiana used for transient transformation was grown in a greenhouse maintained under a 16 h day/8 h night cycle with temperatures of 26 °C and 22 °C, respectively. For the exogenous auxin treatment, leaves of Arabidopsis were sprayed until dripping with 10 μM IAA and were sampled at 0 h, 12 h, 24 h and 48 h.

Cloning of the CgARF1 gene

Total RNA was isolated from plant tissue using a MiniBEST Universal RNA Extraction Kit (Takara, Dalian, China) following the manufacturer’s instructions. The RNA was then reverse transcribed into cDNA using FastKing gDNA Dispelling RT SuperMix (Tiangen, Beijing, China). The CgARF1 gene was amplified with the primer pairs listed in Table 1 and then cloned into the pBI121 vector between two restriction sites (Xba I and Sma I) to construct a recombinant vector. Gene amplification was performed by PCR as follows: 94 °C for 3 min; 35 cycles of 98 °C for 10 s, 60 °C for 15 s, and 72 °C for 30 s; and a final step at 72 °C for 5 min. The vector was then transformed into Trelief™ 5α chemically competent cells (Tsingke, Beijing, China), and the positive clones were tested.

Table 1 List of primer sequences used for the experiments.

Sequence type	Primer name	Primer sequence (5′–3′)	
Gene cloning	CgARF1-F1	GAGAACACGGGGGACTCTAGAATGGCTTTTGCTCCTCTTCATT	
CgARF1-R1	ATAAGGGACTGACCACCCGGGGACATCTTTGTCGGCCTGGTC	
35S-F	GACGCACAATCCCACTATCC	
Fluorescence quantification	CgARF1-F2	GAACCTTCATCCATTTCACGACC	
CgARF1-R2	TGAAGTGGTGGTCTTGCTCGTT	
Subcellular location	CgARF1-F3	ATACACCAAATCGACTCTAGAATGGCTTTTGCTCCTCTTCATT	
CgARF1-R3	TATTTAAATGTCGACCCCGGGGACATCTTTGTCGGCCTGGTC	
1300-F	AACGCTTTACAGCAAGAACGGAATG	
1300-R	TAGGTCAGGGTGGTCACGAGGGT	

Sequence analysis

The conserved domain of CgARF1 was predicted online at the NCBI Conserved Domain Database (https://www.ncbi.nlm.nih.gov/Structure/cdd/wrpsb.cgi) (Lu et al., 2020). ExPASy (https://web.expasy.org/) was used to predict the physicochemical properties and hydrophobic properties of the CgARF1 protein (Gasteiger et al., 2005). Signaling peptides were predicted by using the SignalP 4.0 server (http://www.cbs.dtu.dk/services/SignalP-4.0/) (Petersen et al., 2011). Transmembrane domains were predicted using TMHMM2.0 (http://www.cbs.dtu.dk/services/TMHMM/) (Krogh et al., 2001). Secondary structure prediction was performed using the SOPMA server (https://npsa-prabi.ibcp.fr/cgi-bin/npsa_automat.pl?page=npsa%20_sopma.html). Multiple sequence alignments were analyzed by ClustalX 2.1 software (Larkin et al., 2007). The phylogenetic relationships between CgARF1 and 19 sequences from other species identified previously were analyzed using Mega 6.0 software with the neighbor-joining (NJ) method and a bootstrap value of 1,000 (Kumar, Stecher & Tamura, 2016). The amino acid sequences of the CgARF1 homolog proteins were downloaded from the NCBI website (https://blast.ncbi.nlm.nih.gov/Blast.cgi).

Quantitative real-time PCR analysis

RNA was extracted and reverse transcribed as described above. Quantitative real-time PCR (qRT-PCR) was performed using ChamQ Universal SYBR qPCR Master Mix (Vazyme, Nanjing, China) on a StepOnePlus real-time PCR system (Thermo Fisher, Waltham, MA, USA). The procedure was as follows: 95 °C for 5 min and 40 cycles at 95 °C for 15 s and 60 °C for 1 min. The 18S RNA of C. goeringii was used as the internal reference gene when analyzing the expression levels of CgARF1. The specific primers used for this experiment are shown in Table 1. qRT-PCR data were analyzed using the 2−ΔΔCt method. Three technical replicates of each sample were performed for qRT-PCR.

Subcellular localization assay

A subcellular localization prediction tool, WoLF PSORT, was used to predict subcellular localization (Horton et al., 2007). The ORF of the CgARF1 gene (without a stop codon) was inserted into a modified pCambia1300:GFP vector at double restriction sites (XbaI and SmaI) (Table 1). This vector contains the green fluorescent protein (GFP) reporter gene driven by the CaMV 35S promoter, and allows the expression of recombinant proteins fused to GFP at its N-terminus. Then, the recombinant vector was transformed into Agrobacterium tumefaciens strain GV3101 via the freeze-thaw method (Weigel & Glazabrook, 2005), and the bacteria were then grown in Luria-Bertani (LB) media with both kanamycin and rifampicin at 28 °C. Bacterial cells were harvested by centrifugation and resuspended in an infiltration solution (0.15 mM acetosyringone, 10 mM MgCl2, 10 mM MES, pH of 5.6) at a final OD600 of 0.8. The Agrobacterium suspension mixtures were infiltrated into 6-week-old leaves of N. benthamiana (tobacco) using a needleless syringe (Zhao et al., 2019). Three leaves from each tested plant were injected, and three plants were infected by each gene. The fluorescence signals were observed 48–72 h after injection under a confocal laser scanning microscope (LSM710; Carl Zeiss, Jena, Germany).

Plant transformation in Arabidopsis

pBI121-35S::CgARF1 was transformed into Arabidopsis via the Agrobacterium tumefaciens-mediated floral dipping method (Clough & Bent, 1998). Inflorescences of Arabidopsis were infected for 45–60 s and then incubated in the dark for 20 h. Transgenic seeds were screened on MS media containing 50 mg L−1 kanamycin, and the T3 plants were used for further experiments.

Determination of chlorophyll contents and endogenous phytohormones

Acetone-ethanol solvent extraction was applied for the determination of chlorophyll contents. Arabidopsis leaves collected from the same part of the main stem were soaked in a 1:1 mixture of acetone and ethanol in darkness, and the samples were shaken periodically until the green color completely faded from the leaves. Then, the absorbance values at 663, 646, and 470 nm were determined, and the chlorophyll content was calculated according to Porra’s formula (Porra, 2002). Three biological replicates and three technical replicates of each line were performed for this experiment.

The contents of endogenous hormones in Arabidopsis leaves were determined using enzyme-linked immunosorbent assay (ELISA). Samples were ground in 10 mL of 80% (v/v) methanol extraction medium containing 1 mM butylated hydroxy toluene (BHT) as an antioxidant. The extract was incubated at 4 °C for 4 h and centrifuged at 4,000 rpm for 15 min. The supernatant was passed through C-18 columns and washed with 80% (v/v) methanol, 100% (w/v) methanol, 100% (w/v) ether and 100% (w/v) methanol successively. Then the hormone fractions were dried under N2 and dissolved in phosphate buffer saline (PBS) containing 0.1% (v/v) Tween 20 and 0.1% (w/v) gelatin for analysis.

The monoclonal antigens and antibodies against ABA, GAs (GA1 + GA3) and BR used in ELISA were produced at the Phytohormones Research Institute (China Agricultural University; see He, 1993). ELISA was performed on a 96-well microtitration plate. Each well was coated with 100 μL of coating buffer (1.5 g/L Na2CO3, 2.93 g/L NaHCO3, and 0.02 g/L NaN3) containing 0.25 μg/mL antigens against the hormones, and then incubated for 30 min at 37 °C. After washing four times with PBS containing 0.1% (v/v) Tween 20, each well was filled with 50 μL of sample extracts and 50 μL of 20 μg/mL antibodies, and then incubated and washed as above. Next, 100 μL of color-appearing solution containing 1.5 mg/mL 0-phenylenediamine and 0.008% (v/v) H2O2 was added to each well. The reaction was stopped by 12 mol/L H2SO4 per well. Color development was detected at 490 nm using an ELISA Reader (Model EL310; Bio-TEK, Winooski, VT, USA). Hormone contents were calculated following Weiler, Jordan & Conrad (1981). Three biological replicates of each hormone were performed for this experiment.

IAA treatment of detached leaves of transgenic plants

True leaves from 10-day-old seedlings were cut with scissors on an ultraclean table and then placed on MS medium, MS medium with 0.2 mg·L−1 IAA or MS medium with 0.5 mg·L−1 IAA. On day 12, the rooting rate, the average number of roots and the average length of roots were calculated under a stereomicroscope. The rooting rate was measured ten leaves per time for each line, the average number of roots was measured five leaves per time for each line, and the average length of roots was measure three roots per time for each line. Every measurement was performed for five times independently.

Results

Cloning and sequence analysis of CgARF1

The CgARF1 gene sequence was obtained from C. goeringii transcriptome data. The open reading frame (ORF) of CgARF1 was cloned from the ‘Songmei’ cultivar of C. goeringii; it was 2,049 bp in length and encoded 682 amino acids (Fig. S1A; Table S1). The CgARF1 protein was acidic and exhibited a molecular weight of 76,416.43 and a theoretical isoelectric point of 6.17. This protein exhibits an instability index greater than 40, indicating that it is unstable (Guruprasad, Reddy & Pandit, 1990). Hydrophobicity analysis showed that the CgARF1 protein might be a hydrophobic protein due to its grand average hydropathicity (GRAVY) value of −0.456 (Fig. S1B). No signaling peptides or transmembrane domains were found in this protein (Figs. S1C and S1D). The secondary structure consisted of 19.50% α-helices, 15.25% extended strands, 4.84% β-turns and 60.41% random coil structures (Fig. S1E).

The amino acid sequence of CgARF1 was aligned with those of 19 homologous ARFs from NCBI, and the results showed that they all contained three typical domains: the B3 DNA binding domain, Aux_resp domain and AUX/IXX family domain (Fig. S2). To study the evolutionary relationships between CgARF1 and the 19 ARF transcription factors (TFs) from other species, a phylogenetic tree was constructed (Fig. 1). The results showed that the 20 ARF TFs were divided into three subfamilies, among which CgARF1 showed the closest evolutionary relationships with three ARF7 sequences from the orchids C. sinense, Phalaenopsis equestris and Dendrobium catenatum. This suggests that these ARF genes might have similar biological functions.

Figure 1 Phylogenetic relationships among the amino acid sequences of CgARF1 and 19 ARF genes from other species.

The phylogenetic tree was constructed by MEGA 6.0 software with the neighbor-joining (NJ) method and a bootstrap value of 1,000. The phylogenetic tree was divided into seven subgroups named I, II and III.

Expression pattern of CgARF1 in different tissues and under IAA treatment

Gene expression patterns are typically closely correlated with gene functions. To better understand the function of CgARF1, its expression levels in different tissues (roots, pseudobulbs, leaves, and flowers) of C. goeringii were first examined. The results showed that this ARF gene was expressed in all examined tissues and showed different expression levels in these four organs (Fig. 2A). The highest expression was found in the leaves, while the lowest level was observed in the pseudobulbs (P < 0.05). This suggests that CgARF1 might be involved in leaf development.

Figure 2 Expression pattern of CgARF1 in C. goeringii.

(A) Expression levels of CgARF1 in different tissues. R, root; P, pseudobulb; L, leaf; F, flower. (B) Expression levels of CgARF1 under IAA treatment. The abscissa denotes the time, where 0 h represents the stimulus onset. The significance of differences was estimated using ANOVA and Duncan’s tests. Different letters indicate significant differences (P < 0.05).

As an auxin response factor, CgARF1 responded strongly to exogenously supplied auxin (IAA), as shown in Fig. 2B. The expression level of CgARF1 sharply increased after spraying with IAA and peaked at 2 h after treatment, while its expression at 12 h was similar to that at 2 h. A significant decrease was observed after 12 h. This up-down-up-down expression pattern indicates the complexity of the regulatory relationship between CgARF1 and exogenous auxin.

Subcellular localization of the CgARF1 protein

To investigate the function of a gene at the protein level, it is important to determine its possible site of residency in the cell. The online analysis tool WoLF PSORT predicted that the CgARF1 protein showed the greatest probability of being located in the nucleus. To determine the subcellular localization of CgARF1, we constructed a CgARF1::GFP fusion protein. Through Agrobacterium-mediated infiltration, this fusion protein was transiently expressed in N. benthamiana. Under a laser confocal microscope, the nuclear DNA was stained with fluorescent blue DAPI. CgARF1::GFP fluorescence largely overlapped with the blue fluorescence, indicating that the CgARF1 protein was mainly located in the nucleus (Fig. 3).

Figure 3 Subcellular localization of the CgARF1 protein in Nicotiana benthamiana epidermal cells.

Identification and phenotypic observation of transgenic plants overexpressing CgARF1

To confirm the overexpression of the CgARF1 gene in transgenic plants, total genomic DNA (gDNA) was extracted from the mature leaves of Arabidopsis. PCR detection was conducted with the 35S forward primer and the CgARF1 reverse primer using wild-type (WT) DNA as the negative control. As shown in Fig. S3, an approximately 2,000 bp product was amplified from transgenic Arabidopsis, while no target PCR product was amplified from WT. Quantitative real-time PCR results also showed that the expression level of the CgARF1 gene in the transgenic lines was much higher than that in the wild type (Fig. 4).

Figure 4 The expression level of overexpression CgARF1 transgenic plants and WT plants.

Different numbers after the gene name represent different lines of transgenic Arabidopsis. The significance of differences was estimated using ANOVA and Duncan’s tests.

Three independent transgenic lines from the T3 generation were randomly selected for phenotypic observation. By observing the entire growth cycle of Arabidopsis, it was found that the CgARF1-overexpressing transgenic plants mainly displayed earlier flowering (Fig. 5A) and earlier senescence of rosette leaves (Figs. 5B and 5C) than WT plants. On day 40 after transplantation into soil, the leaf shape of the transgenic plants showed no change, but leaf etiolation was obvious. In response to this phenomenon, the chlorophyll contents were measured. The results showed that the overexpression of CgARF1 had significant effects on the contents of chlorophyll a, chlorophyll b and total chlorophyll (Figs. 5D–5F). However, there was no obvious effect on chlorophyll a/b between the transgenic and wild-type plants (Fig. 5G).

Figure 5 Phenotypic observation of T3 CgARF1-overexpressing Arabidopsis plants.

(A) Side view of WT and CgARF1 transgenic plants grown for 30 days in soil. (B) Top view of WT and CgARF1 transgenic plants grown for 40 days on soil. (C) Leaf phenotypes of WT and CgARF1 transgenic plants. (D–G) Chlorophyll contents of leaves from WT and CgARF1 transgenic plants grown for 40 days in soil. The significance of differences was estimated using ANOVA and Duncan’s tests. Different letters indicate significant differences (P < 0.05).

Growth of detached leaves under IAA treatment

Individual leaves from WT and overexpressing CgARF1 Arabidopsis plants were excised and inoculated onto MS medium supplemented with different concentrations of IAA. On the hormone-free MS medium, wild-type leaves developed symptoms of yellowing and wilting, whereas most of the detached leaves from transgenic plants remained green (Fig. 6A). With an increase in the concentration of hormones, transgenic leaves gradually showed an upward curling phenotype (Fig. 6B). Additionally, their rooting rate, rooting number and rooting length were all obviously increased compared with those of leaves without hormone treatment (Figs. 6C–6E). However, the WT leaves still grew poorly and slowly.

Figure 6 Phenotypic observation of detached leaves from T3 CgARF1-overexpressing Arabidopsis plants under IAA treatment.

(A) Growth status of detached leaves from WT (left) and CgARF1 transgenic (right) plants on MS medium. (B) Observation of rooting from detached leaves of WT (left) and CgARF1 transgenic (right) plants under a stereomicroscope. (C) Rooting analysis of detached leaves from WT and CgARF1 transgenic plants. The significance of differences was estimated using ANOVA and Duncan’s tests. Different letters indicate significant differences between two samples within a single group (P < 0.05).

Changes in endogenous phytohormone contents under IAA treatment

The contents of endogenous hormones, including IAA, ABA, GAs, and BR, were also detected in the mature leaves of Arabidopsis (Fig. 7). The results showed that the hormone levels in transgenic Arabidopsis and WT showed the same trend under IAA treatment, but the levels in the former were almost lower than those in the latter, especially for GAs. In addition, at the later stage of IAA treatment, the content of GAs in transgenic plants increased significantly after 24 h, while the ABA contents also showed similar changes after 12 h.

Figure 7 Contents of endogenous hormones in T3 CgARF1-overexpressing Arabidopsis plants under IAA treatment.

The abscissa denotes the time, where 0 h represents the stimulus onset. The significance of differences was estimated using ANOVA and Duncan’s tests. Different letters indicate significant differences between two samples within a single group (P < 0.05).

Discussion

ARF transcription factors are a critical component of the auxin response system, which plays an important role in regulating plant growth and development (Guilfoyle & Hagen, 2007). Orchids are traditional ornamental plants, and their cultivation and conservation are therefore always matters of considerable concern. Leaves are a basic vegetative organ in plant development and productivity because they are capable of photosynthesis. The phylogenetic tree generated in this work showed that CgARF1 presents a close relationship with several ARF7 genes of Orchidaceae. Among these genes, CsARF7 has been reported to be associated with leaf color variation (Zhu et al., 2015). Among the ARFs of Arabidopsis, CgARF1 has the highest homology to AtARF1. Previous research revealed that AtARF1, together with AtARF2, regulated the senescence of rosette leaves (Ellis et al., 2005). These findings all have some implications for this study. In this study, we first found that the expression level of CgARF1 was higher in leaves than in other tissues, which provides a potential basis for functional studies of this gene. Subsequently, rosette senescence was observed in CgARF1-overexpressing Arabidopsis. One of the most evident features of plant senescence is the gradation of chlorophyll (Lim, Kim & Nam, 2007). Therefore, we determined the chlorophyll contents of leaves from wild-type and transgenic Arabidopsis plants to obtain further evidence supporting the conclusion that CgARF1 positively regulates rosette leaf senescence. However, there were no obvious phenotypic changes in leaf morphology or leaf color, indicating that CgARF1 may not regulate these processes by itself.

Generally, the expression of ARF genes is directly regulated by auxin (Wang et al., 2018). In this study, the expression level of CgARF1 was elevated at 2 h and 12 h after IAA treatment and was then significantly decreased at 24 h. This expression pattern showed that CgARF1 was strongly induced by auxin and that the underlying mechanism of action is complicated. In addition, detached leaves of wild-type and transgenic plants were inoculated on MS medium without and with varying concentrations of IAA to compare their rooting phenotypes. We found that both the wild-type and transgenic leaves showed rooting on all media, which is a common mechanism of tissue self-repair when plants suffer from external damage (Xu & Huang, 2014; Kareem et al., 2016). However, the average number of roots, length of roots and rooting rate of transgenic plants were significantly increased compared with those of WT. The higher the concentration of exogenous IAA was, the more roots that were produced. In view of the extensive research on the regulation of de novo organogenesis in plants in recent decades, phytohormones are considered to be the critical factors affecting this process (Sangwan, Sangwan-Norreel & Harada, 1997; Ikeychi, Sugimoto & Iwase, 2013; Liu et al., 2014). The regeneration of adventitious roots is strongly induced by auxin, and the production, transport and signaling of auxin may all be involved (de Klerk, van der Krieken & de Jong, 1999). Together with previous studies, our results show that rooting of detached leaves is due to the production of free auxin induced by wounding. Free auxin is then highly concentrated in procambium stem cells and their surrounding parenchyma cells via polar auxin transport, resulting in their transformation into root founder cells, from which adventitious roots are produced (Liu et al., 2014). Our results showed the clear involvement of the CgARF1 gene in this process. We speculate that elevated concentrations of auxin in detached leaves induced CgARF1 expression, which produced a series of auxin response processes, including the formation of adventitious roots caused by the transformation of cell roles. Therefore, the gene expression pattern relies on the auxin distribution, and auxin induction depends on the wounding of leaves at the beginning of de novo root organogenesis.

Numerous studies on plant physiology have shown that there is cross-talk among plant hormones (Woodward & Bonnie, 2005). When the auxin level is high, auxin promotes the ubiquitination and degradation of Aux/IAAs through a SCFTIR1/AFB-proteasome module, and releases ARF proteins (Wang & Estelle, 2014), in turn regulating the levels of other endogenous hormones. Wang et al. (2011) found that ARF2 is a novel regulator in the ABA signal pathway, which has crosstalk with auxin signal pathway in regulating plant growth. In addition, auxin can stimulate DWARF4 expression and BR biosynthesis in Arabidopsis. ARF7 is involved in this process by binding to the promoter of DWARF4 (Chung et al., 2011). In this study, altered GAs, ABA and BR levels indicated that the overexpression of CgARF1 can also modify the contents of endogenous hormones in leaves, and they changed with the treatment of IAA as well, which was similar to the results of previous study. However, the specific mechanism of action between them remains to be explored.

Conclusions

In this study, the coding sequence of the CgARF1 gene was cloned from C. goeringii ‘Songmei’ and functionally verified by overexpression in Arabidopsis. CgARF1-overexpressing lines exhibited leaf senescence, an increased rooting number from detached leaves and changes in endogenous hormone levels under IAA treatment. This study provides useful information and increases our understanding of the regulatory mechanisms of CgARF1 in leaf development during the IAA response.

Supplemental Information

Supplemental Information 1 Raw data.

Click here for additional data file.

Supplemental Information 2 CgARF1 sequences.

Click here for additional data file.

Supplemental Information 3 Clonging and sequence analysis of CgARF1.

(A) Electrophoresis image of PCR amplification. M: DL2000 Marker; 1: CgARF1. (B) Hydrophobic properties prediction by ExPASy. (C) Signal peptides prediction by SignalP 4.0. (D) Transmembrane domains prediction by TMHMM2.0. (E) Secondary structure prediction by SOPMA.

Click here for additional data file.

Supplemental Information 4 Sequence alignment among amino acid sequences of CgARF1 and 19 ARF genes from other species.

Click here for additional data file.

Supplemental Information 5 The PCR identification of overexpression CgARF1 transgenic plants.

M: DL2000 Marker; CK-: the test sample of WT. A1-A9: test samples of transgenic lines.

Click here for additional data file.

Additional Information and Declarations

Competing Interests

Author Contributions

DNA Deposition

Data Availability

The authors declare that they have no competing interests.

Zihan Xu performed the experiments, analyzed the data, prepared figures and/or tables, authored or reviewed drafts of the paper, and approved the final draft.

Fangle Li performed the experiments, analyzed the data, prepared figures and/or tables, authored or reviewed drafts of the paper, and approved the final draft.

Meng Li analyzed the data, prepared figures and/or tables, and approved the final draft.

Yuanhao He analyzed the data, prepared figures and/or tables, and approved the final draft.

Yue Chen conceived and designed the experiments, authored or reviewed drafts of the paper, and approved the final draft.

Fengrong Hu conceived and designed the experiments, authored or reviewed drafts of the paper, and approved the final draft.

The following information was supplied regarding the deposition of DNA sequences:

The sequences are available in the Supplemental Files and at GenBank: OL412398.

The following information was supplied regarding data availability:

The raw measurements are available in the Supplemental File.

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
