# Peer review of "Functional analysis of ARF1 from Cymbidium goeringii in IAA response during leaf development"

_PeerJ, doi:10.7717/peerj.13077_

## Round 0.1 · original submission · Major Revisions

I have received the decision from three Reviewers on your manuscript entitled "Functional analysis of ARF1 from Cymbidium goeringii in IAA response during leaf development".

All reviewers requested reformatting figures and text, with a clear representation of methods, figure legends, and interpretation of the data. Explanation of the relationship between ARFs and leaf development is also necessary. The authors should reply to all of the concerns from reviewers. In addition, English editing must be required for the revised version.

Reviewer 1 ·

Basic reporting

1. The English language was so bad, and it should be improved to avoid Chinglish-expression forms and grammatical errors in the whole manuscript. I suggest that the author should have a colleague, who is proficient in English and familiar with the and familiar with the subject matter, to review your manuscript, or contact a professional editing service to revise it.

2. I have several criticisms to this manuscript for the messy format of the papers, such as the “left-aligned text”, the “random space” in the line 55.

3. There exist some mistakes in manuscript, such as the figure legends of Fig. 2, which does not match up with the Fig.2A.

4. The authors were not able to describe these results successfully, besides, they also could not explain about the cross-talk between ARFs and leaf development

Experimental design

No

Validity of the findings

No

Additional comments

No

Reviewer 2 ·

Basic reporting

This paper demonstrated the cloning of ARF1 from Cymbidium goeringii and ectopic overexpression of CgARF1 in Arabidopsis plants. I have many concerns on Figures and methods. Especially, essential data, and information are missing in legends and main text.

1) The description of Figure legend should be improved in all figures. The essential informations were not mentioned in Figure legends. For example, In, Fig. 5A and 5B, which group represents WT or transgenic plants?

2) Figure legends are missing in all supplemental Figures.
Line 207-209 Authors compared CgARF1 and 19ARFs (Fig. 1 and S2). However,the plant species and gene/protein ID were missing in Figure 1 legend.

3) Statistical analysis procedures are not shown in Figure. Which tests is applied to data, Post-hoc test, Tukey-Kramer test, Bonferroni test, Dunnet test etc, please also specify the sample number in each figure panel. Please specify the number of biological replicates and technical replicates in the methods section or legends..

4) Bar graphs of Fig. 2A and 2B are incorrectly displayed. Fig. 2A and 2B should be exchanged each other.

5) In the methods section, Line 162
The ORF of the CgARF1 gene (without a stop codon) was inserted into the pCambia1300 vector (Table 1), which contains the green fluorescent protein (GFP) reporter gene driven by the CaMV 35S promoter. Please describe the correct methods of plasmid construction. Thus, the control of GFP vector was not indicated. In the text (line 231), CgARF1::GFP fusion protein was described. The position of GFP fused with CgARF1 protein (N / C terminus) was not specified according to the method section (pCambia1300 vector).

6) pCambia1300 vector does not contain GFP under 35S promoter.
https://www.snapgene.com/resources/plasmid-files/?set=plant_vectors&plasmid=pCAMBIA1300

7) Line 70 However, all ARF proteins do contain all three complete domains, as is the case for AtARF3/7/13/23 in Arabidopsis (Guilfoyle et al., 2001; Remington et al., 2004) and CiARF3/14/17 in sweet orange (Li et al., 2015).

“all ARF proteins do contain all three complete domains” is correct? Probably, all ARF proteins do NOT contain?

8) “as is the case for AtARF3/7/13/23 in Arabidopsis” is not correct. AtARF7 contains three complete domains. Probably, ARF7 >ARF17.

9) Line 77, AtARF7 and AtARF9 are also induced in senescent leaves (Lin and Wu, 2004). This sentence would not be appropriate, because the cited paper did not describe the ARF expression in the text although raw data might contain the expression of ARF7 and ARF9.

10) Line173, Author constructed pBI121-35S::CgARF1. pBI121 is 35S::ORF-GUS vector. Is the CgARF1 protein expressed as GUS-fusion?

https://www.snapgene.com/resources/plasmid-files/?set=plant_vectors&plasmid=pBI121

11) Line 191 MS medium with 0.2 mg·L-1 or MS medium with 0.5 mg·L-1. > MS medium with 0.2 mg·L-1 IAA or MS medium with 0.5 mg·L-1 IAA

12) Line 238-241
The CgARF1 overexpression in transgenic lines should be confirmed by qRT-PCR. The genomic PCR analysis did not show any evidence of CgARF1 expression in transgenic lines

13) Line 260 Changes in endogenous phytohormone contents under IAA treatment

13-1) Please describe full details of phytohormone measurement by ELISA. Please indicate the commercial vendor of mAb for each hormone or ELISA kit if used (code number and vendor) .

13-2) Line 264, After 12 h of treatment, exogenous IAA treatment caused a substantial increase in endogenous IAA levels. The IAA level in CgARF1-overexpressing plants began to decline after 12 h, but the level in WT plants remained high.

> I could not understand why IAA level is measured after exogenous IAA treatment. exogenous IAA and endogenous IAA could not be distinguished by ELISA methods. If synthetic auxins NAA or 2,4-D are used, IAA might be measured by ELISA. However, exogenous IAA was used in this experiment. The authors would like to examine the rate of IAA inactivation between WT and CgARF1-overexpressing lines? Discussion (lines 319-323) should be improved.

13-3) Line 261 and 266, The contents of endogenous hormones, including IAA, ABA, GA3, and BR,
Fig. 6B-D shows that the contents of other endogenous hormones in transgenic plants were almost all lower than those of WT plants, especially for GA3.

>Plant does not produce GA3 as endogenous active GA, GA1, and GA4 are endogenous gibberellins, GA3 is produced by the fungus.

13-4) Line 264 exogenous IAA treatment caused a substantial increase in endogenous IAA levels.
>Please indicate the method for exogenous IAA treatment and the concentration of IAA.

Experimental design

Experiments seem to be well designed and performed, but many errors are documented in the method section. The expression of CgARF1 was not confirmed by qRT-PCR. The insertion of CgARF1 gene into the Arabidopsis genome was confirmed by genomic PCR. This qRT-PCR data is crucial for the confirmation of CgARF1 overexpression.

Validity of the findings

I could not confirm all data in this study as the legends of supplemental Figures are missing.

Reviewer 3 ·

Basic reporting

This article studied ARF1 from Cymbidium goeringii. Researchers analyzed basic information of CgARF1 using straightforward bioinformatic tools and identified CgARF1 expression pattern in WT and IAA treatment. More important, through overexpression of CgARF1 in Arabidopsis, authors investigated the potential function of CgARF1 during leaf development under IAA treatment compared with WT plants. Overall, conclusions of this article are appropriately supported by well-designed experiments and comprehensive analysis.

Experimental design

Figure 2 is supposed to show the expression level of CgARF1 in different tissues during the time course and under IAA treatment as well. However, the data shown in Figure 2 could not reflect or support the explanation/ conclusion in Line 215-226. Please provide elaborate description of this Figure 2 regarding but not limit to the following concerns,
a. which organ (R, P, L, F) is the expression level shown in Figure2A from?
b. As indicated in Line 224, where is the data in Figure 2B showing 2h, 12h expression?
c. It would be more supportive to include qRT-PCR raw data in supplementary set.

Validity of the findings

no comment

Additional comments

no comment

---

## Round 0.2 · accepted · Accept

The revised version has been improved by answering concerns from three experts. All reviewers evaluated the suitability of the manuscript for publication in PeerJ.

Reviewer 1 ·

Basic reporting

The revised manuscript has been much improved and answered all my concerns. I have no more comments.

Experimental design

The revised manuscript has been much improved and answered all my concerns. I have no more comments.

Validity of the findings

The revised manuscript has been much improved and answered all my concerns. I have no more comments.

Additional comments

The revised manuscript has been much improved and answered all my concerns. I have no more comments.

Reviewer 2 ·

Basic reporting

The revised manuscript is highly improved and satisfied all my concerns. I think that this round manuscript is suitable for publication in the PeerJ article.

Experimental design

The experimental design is appropriate to support the conclusion.

Validity of the findings

All provided data support the conclusion.

Reviewer 3 ·

Basic reporting

In the revised manuscript, authors resolved all concerns from my previous review. This article studied the properties and function of CgARF1 from C. goeringii. This study provides useful information as a preliminary investigation of ARF1 roles in leaf development. The underlying mechanisms still needs to be explored.

Experimental design

no comment

Validity of the findings

no comment

Additional comments

no comment